# Relationships between Inflammation and Age-Related Neurocognitive Changes

**DOI:** 10.3390/ijms232012573

**Published:** 2022-10-20

**Authors:** Run Jin, Aidan Kai Yeung Chan, Jingsong Wu, Tatia Mei Chun Lee

**Affiliations:** 1State Key Laboratory of Brain and Cognitive Sciences, The University of Hong Kong, Hong Kong 999077, China; 2Laboratory of Neuropsychology and Human Neuroscience, The University of Hong Kong, Hong Kong 999077, China; 3Faculty of Medicine, The University of Hong Kong, Hong Kong 999077, China; 4College of Rehabilitation Medicine, Fujian University of Traditional Chinese Medicine, Fuzhou 350108, China

**Keywords:** inflammation, aging, cognition, neurodegeneration, brain

## Abstract

The relationship between inflammation and age-related neurocognitive changes is significant, which may relate to the age-related immune dysfunctions characterized by the senescence of immune cells and elevated inflammatory markers in the peripheral circulation and the central nervous system. In this review, we discuss the potential mechanisms, including the development of vascular inflammation, neuroinflammation, organelle dysfunctions, abnormal cholesterol metabolism, and glymphatic dysfunctions as well as the role that the key molecules play in the immune-cognition interplay. We propose potential therapeutic pharmacological and behavioral strategies for ameliorating age-related neurocognitive changes associated with inflammation. Further research to decipher the multidimensional roles of chronic inflammation in normal and pathological aging processes will help unfold the pathophysiological mechanisms underpinning neurocognitive disorders. The insight gained will lay the path for developing cost-effective preventative measures and the buffering or delaying of age-related neurocognitive decline.

## 1. Introduction

The aging population is a worldwide trend [1]. Data from the World Health Organization indicate that by 2030, 1.4 billion people will be 60 or older, and this figure will double by 2050. Among various psychological and physiological changes associated with aging, changes in immune functioning have attracted much research attention due to the way they affect neurocognitive aging [2,3,4].

## 2. Inflammation and Cognitive Aging

Chronic inflammation is a form of slight and long-lasting inflammation that lasts for several months or years and that is especially prevalent among older people [5,6]. Sources of inflammation include the accumulation of oxidative stress, mitochondrial dysfunction, cell senescence, and the general weakening of the immune functions [6]. At the cellular level, aging is accompanied by a decrease in mitochondrial efficiency, leading to excessive reactive oxygen species (ROS) production, a byproduct of oxygen metabolism and maintaining adequate production of adenosine triphosphate (ATP). Moreover, the mitochondrial DNA (mtDNA) located in the mitochondria is susceptible to DNA damage due to the lack of protection from nucleosomes [7]. During aging, increased ROS production triggers DNA damage, cell apoptosis, and necrosis and causes inflammation. At the physiological level, reduced naïve T and B cells and increased memory cells are circulated through the blood [8]. Meanwhile, a shift occurs between Th1 (which produces interferon (IFN)-γ and interleukin (IL)-2) and Th2 (which produces IL-4 and IL-10) cells that results in an increased Th1/Th2 ratio and a proinflammatory environment as age increases [9]. Overall, these changes may induce and/or exacerbate inflammation.

In recent years, increasing research has shown that dysfunctional immune systems might underpin neurocognitive changes associated with normal and pathological aging [10,11,12]. In animal studies, systemic inflammation could induce cognitive impairment measured by spatial-learning tasks in mice [13]. In healthy human participants, serum interleukin-6 (IL-6) increased with age and acted as a mediator partially explaining the relationship between chronological age and impairments in processing speed [14].

Previous studies demonstrated that there were elevated inflammatory markers in people with Alzheimer’s disease (AD) or mild cognitive impairment (MCI) [15]. In addition, participants with AD showed a high level of neuroinflammation compared to healthy controls in frontotemporal regions [16], which involve several aspects of cognitive functions, including attention and working memory. Previous research has also revealed that genetic susceptibility and chronic inflammation interaction in people with a high level of C-reactive protein (CRP; ≤ 8 mg/L), an indicator of inflammation states secreted in response to cytokines that indicates an increased risk of AD and of earlier disease onset [17].

In the brain, microglial activation, the process of neuroinflammation [18], could be triggered by amyloid peptides, fibrils, and amyloid precursor protein (APP), which in turn help clean the pathological proteins in neurodegeneration [19]. However, it is speculated that the prolonged activation of microglia might eventually become uncontrollable and dysfunctional, releasing excessive inflammatory cytokines and chemokines and showing attenuated phagocytosis capacity, causing neuron death and resulting in cognitive impairment [20,21]. Thus, aging is related to chronic inflammation. Meanwhile, the elevation of inflammatory markers is associated with neurodegenerative changes. An overview of the mechanisms underlying inflammation and neurocognitive decline is shown in Figure 1. 

## 3. Key Molecules Bridging Inflammation and Cognition

Cytokines are a complex network of soluble polypeptides or glycoproteins delivering inter- and intracellular messages by binding to various types and forms of receptors (i.e., cell-surface, membrane-anchored, or soluble form). Immune cells can synthesize and secrete cytokines via autocrine, paracrine, and endocrine functions. Cytokines can be classified into five families (e.g., interleukins, interferons, tumor necrosis factor (TNF), growth factors, and chemokines) and further divided into small subgroups under each family [22]. At the molecular level, chronic inflammation is characterized by increased proinflammatory cytokines with increased age. Many studies have shown that overall levels of inflammatory markers are higher in older people without acute infections [5,23,24]. Among these markers, IL-6, IL-8, IL-2, IFN-γ, and TNF-α have shown reliable elevation with advanced age [25].

In the central nervous system (CNS), cytokines support and modulate various cognitive functions (e.g., memory consolidation and learning) by playing roles in balancing neuronal vitality, such as maintaining synaptic plasticity and synaptic scaling, adjusting long-term potentiation, and regulating neurogenesis. They also play a role in the removal of harmful chemicals and act as ubiquitous and indispensable mediators in neuroinflammation and pathological protein clearance and accumulation [26,27,28,29,30,31]. As age-related neurodegenerations are highly attributed to increased neuronal vulnerability and protein aggregation during accelerated brain-aging processes [32], the effect of alterations in the concentrations of cytokines on neurocognitive decline is evident. Abnormal levels of inflammatory markers have been reported in people with AD and MCI. One meta-analysis [33] including more than 10,000 AD patients and healthy controls revealed that IL-1β; IL-2; IL-6; IL-18; IFN-γ; high-sensitivity CRP; C-X-C motif chemokine-10, a TNF-α converting enzyme; and soluble TNF receptors 1 and 2 are elevated in the peripheral system. In short, we speculate that age-related immune senescence and chronic inflammation lead to changes in cytokine levels in the periphery and in the brain globally or locally, which are jointly associated with neurodegenerative changes and related to behavioral and symptomatological manifestations. Here, we mainly focus on IL-6, IL-12, IL-1β, IL-18, and IFN-γ as key molecular connectors between chronic inflammation and cognition given their central roles in immunological pathways and typical roles in the regulation of neural activities and environmental homeostasis in the CNS. More specific discussions on the roles of other immune-related biomarkers, including interferons [34], chemokines [35], and TNF [36], as well as cytokine networks [37] in the CNS, are given elsewhere.

### 3.1. IL-6

IL-6 is a versatile cytokine in promoting inflammatory response and protecting homeostasis by inducing the synthesis of acute-phase proteins (i.e., CRP) and stimulating acquired immunity [27,38]. Studies have also demonstrated its modifying role in age-related chronic inflammation. Animal research proved that IL-6 has a regulatory role in cytokine balance during the aging process given that among IL-6-knocked-out mice, older mice will release more proinflammatory cytokines after lipopolysaccharide stimulation whereas, in wide-type mice, the interaction’s direction is the opposite [39]. Human and animal studies have shown that elevated peripheral levels of IL-6 are related to cognitive impairment. In humans, an inverse relationship exists between IL-6 levels and global cognitive status (measured by Mini-Mental State Examination (MMSE) scores) [33]. Longitudinal data also verified that people with higher IL-6 circulation were 1.42 times more likely to experience global cognitive decline after 2 to 7 years of follow-up than those with lower IL-6 levels [40]. People with higher IL-6 and IL-10 levels are also highly likely to be diagnosed with MCI over time [41]. Mechanistically, IL-6 levels influence neurocognitive functions by exacerbating neuroinflammation, controlling adult neurogenesis, and provoking the deposition of amyloid beta. For instance, a randomized controlled trial included qigong intervention to promote cognitive functions in older people [42]. The results revealed that the decrease in peripheral IL-6 and its modulating effect on hippocampus volume changes explain the qigong exercise’s positive effect on processing speed. In animal studies, the potential mechanisms of the effect of IL-6 have also been explored. A study observed the IL-6 pathway activation in the hypothalamus and hippocampus in AD model mice, while inhibiting the signal transducer and activator of transcription 3 (STAT3) ameliorates memory impairment and reduces plasma IL-6 levels [43]. In transgenic mice that ultimately overexpress human APP and develop AD type neuropathology, elevated IL-6 mRNA expression in the hippocampus and cortex before the formation of amyloid plaques was observed [44]. Moreover, the presence of the soluble IL-6 receptor (sIL-6R) and IL-6 together triggers the production of cell-associated and secreted forms of APP [45], which is closely associated with neurocognitive impairment and the development of AD.

### 3.2. IL-12

IL-12, or IL-12p70, is a 70 kDa heterodimeric cytokine comprising the subunits p35 and p40 linked covalently. Two separate genes, IL-12A (p35) and IL-12B (p40), encode the subunits [46]. By binding to its heterodimeric receptor formed by IL-12Rβ1 and IL-12Rβ2, IL-12 promotes an inflammatory response by triggering the Janus kinase (JAK) signal transducer and activator of transcription (STAT) signaling pathway and bridges innate and adaptive immunity [47]. In the periphery, IL-12 from monocytes induces IFN-γ production from Th1 cells, which subsequently leads to the M1 activation of monocytes to defend against infection [48]. It also participates in the development of CNS autoimmune diseases, such as multiple sclerosis, which implies its potential role in sustaining chronic neuroinflammation [49]. Notably, its functions rely highly on its structural basis because the dimerization of α subunits (e.g., IL-12p35) and β subunits (e.g., IL-12p40) stimulates chronic inflammatory diseases whereas the combinations of IL-12p35 and other β subunits Ebi3 (e.g., compositions of IL-35 or IL-27) inhibit inflammation and relieve autoimmune diseases. Yet, the relationship between IL-12 and cognitive functions is inconsistent in human studies. A series of IL-12-associated genes is related to cognitive aging [50]. Alternatively, AD and MCI patients with increased IL-10 and IL-12 had larger hippocampal volumes, more than 42 amino acid forms of amyloid β (Aβ1-42) in their cerebrospinal fluid (CSF), and less phosphorylated tau, implicating the protective role of both ILs [51]. To complicate the situation further, IL-12 showed protective and harmful effects in the CNS. IL-12p35, the alpha subunit of the IL-12 or IL-35 cytokine, inhibits the expansion of pathogenic Th17 and Th1 cells and inhibits cytokine-induced activation of STAT1 and STAT3 pathways in the mouse model of human multiple sclerosis [52], supporting its advantageous effect. Furthermore, higher IL-12p70 is associated with slower cognitive decline as well as less tau and neurodegeneration in participants with higher Aβ [53]. In contrast, in the APP/PS1 Alzheimer’s disease mouse model, microglia increase production of the IL-12 and IL-23 subunit p40, and neutralization of p40 by antibodies reduces the cerebral amyloid load [54]. Overall, we can possibly attribute the discrepancy to the effect of the isolated subunits versus the bioactive dimeric forms. In other words, IL-12 may protect and harm the aging brain via distinct cellular mechanisms, depending on the involvement of specific subunits.

### 3.3. IL-1β and IL-18

As members of the IL-1 family of cytokines, IL-1β and IL-18 are prominent mediators linking chronic inflammation and neurodegeneration. Elevated IL-18 in the peripheral circulation is consistently related to worse cognitive performance in several groups of people. The IL-18 levels show a peak during mild cognitive impairment compared to healthy controls and AD patients [55]. In first-episode psychosis patients, IL-18 levels are positively related to cognitive impairment [56]. Furthermore, mediation analysis has suggested that IL-1β and IL-18 partially mediate the relationship between vitamin D/25(OH)D3 deficiency and the risk of cognitive impairment in older people [57]. The potential mechanisms include stimulating inflammatory responses and modulating long-term potentiation (LTP). Specifically, they are involved in one of the central processes in chronic inflammation: NLRP3 inflammasome activation [58]. As an intracellular multiprotein complex initiating innate immune responses, the inflammasome would be easily triggered once it recognizes the pathogen-associated and damage-associated molecular patterns, which are usually elevated with aging. It will then increase the activity of cleaved caspase-1 (CC1) and -8, cleaving the precursors of proinflammatory cytokines (e.g., IL-1β and IL-18) and then producing mature forms [59]. Furthermore, in NLRP3-deficient APP/PS1 mice, reduced Aβ deposition, anti-inflammatory phenotypes of microglia, and reversed loss of spatial memory were observed, implying that suppression of the inflammatory response indeed reverses neurocognitive changes [60]. At the same time, IL-1β and IL-18 also mediate the accomplishment of long-term potentiation, a process of long-lasting strengthening of synaptic efficacy fundamentally underlying learning, memory, and motivation in mice. Studies have shown that IL-18 can attenuate LTP in rat dentate gyrus slices in vitro (e.g., [61]). Nevertheless, another study unveiled that TNF-α and IL-1β but not IL-18 can suppress chemically induced LTP and brain-derived neurotrophic factor signaling in isolated hippocampal synaptosomes of mice [62]. The differences in the experimental tissues and the discrepancy in results suggest that IL-1β affects neurons and synaptic plasticity directly and that the effect of IL-18 might be an indirect result of the synergistic action of multiple cells and pathways. Considering that aged rats showed impaired LTP functions in the hippocampus [63,64] and that oligomeric Aβ peptides inhibited hippocampal LTP [65], the connections are strong between the IL-1 family of cytokines, chronic inflammation, and deficits in neurocognitive functions during aging.

### 3.4. Interferons

In addition to interleukins, interferon families are remarkable messengers in the CNS. The term *interferon* refers to the property of interfering with viruses and bacteria [66]. Interferons come in three types: type I, which includes IFN-α and IFN-β; type II (IFN-γ); and type III (IFN-λ or interleukin-28/29). IFN-γ is a proinflammatory cytokine encoded by the *IFNG* gene. The structure of monomers is a core of six α-helices and an unfolded sequence in the C-terminal, which are linked to form the biologically active dimer. It also triggers the immune responses through a JAK-STAT signaling pathway [67]. Changes in the levels of interferons in healthy cognitive aging and MCI or AD patients have not been widely reported [68], and the directionality is inconsistent. It is necessary to scrutinize their roles in neurocognitive decline given their unique immunological properties and compelling clues at the micro levels. It is speculated that IFN-γ mediates chronic inflammation and neurodegeneration via several mechanisms, including the involvement of inflammatory responses and immune surveillance as well as the effects on neurogenesis and Aβ digestion. Researchers have highlighted the dual effects of the type II interferon—IFN-γ—in immune-mediated demyelinating disorders, such as multiple sclerosis and experimental autoimmune encephalomyelitis [69,70]. Meanwhile, Deczkowska and colleagues [71] proposed that during the aging of the brain, an imbalance occurs between type I and type II interferons, with excessive type I IFNs and insufficient IFN-γ signaling at the choroid plexus in controlling the entrance of leukocytes from the periphery to the CSF. Therefore, these age-related decreases in beneficial immune surveillance are associated with subsequent cognitive decline. Additionally, the effect of IFN-γ is implied in the disruption of neural-cell genesis and synaptic plasticity, activation of glial cells, and dampening or relief of the Aβ burden [28]. More specifically, IFN-γ injection in animals primes microglia with morphological changes and leads to the release of proinflammatory cytokines (IL-1β, TNF-α, and IL-6) and nitric oxide, particularly a reduction in hippocampal neurogenesis [72]. Conversely, inhibition of the JAK-STAT pathway could suppress neuroinflammation processes. Moreover, IFN-γ might also influence cognition through its interaction with APP [73], yet the evidence is conflicting. In APP/PS1 mice, IFN-γ secreted by the infiltration of T cells in the brain promotes microglial activation, and anti-IFN-γ antibodies not only attenuated Aβ deposition induced by CD4(+) T but also attenuated impaired cognition [74]. However, other studies have also shown that intraperitoneal administration of IFN-γ significantly reduced the burden of Aβ plaque in the cortex and hippocampus and even protected against cognitive deficits [75]. One possible explanation is that IFN-γ is inherently pleiotropic. Another reason may be the methodological differences in the dose, disease stage, or experimental manipulations. In sum, the multifaceted nature of IFN-γ and the relationship between IFN-γ and cognition in humans require further attention and delineation.

## 4. Neuroinflammation

Contrary to the traditional view that the brain is an immune-privileged area, it is now well recognized that resident brain cells (i.e., microglia, astrocytes, and oligodendrocytes) are actively involved in defense and clearance processes [76,77]. Previous studies have shown that neuroinflammation is related to age-related neurocognitive changes and AD [78,79]. One can assess the levels of neuroinflammation with the measuring translocator protein-18 kDa (TSPO), a transmembrane-domain protein located on the outer mitochondrial membrane. During microglial and astrocyte activation, its expression levels increase drastically compared to normal conditions [80]. Relative to their healthy counterparts, people with AD or MCI showed a significant increase in TSPO levels at the whole-brain level, especially within the frontotemporal regions. In addition, research uncovered that there was a negative association between TSPO levels in the parietal lobe and MMSE scores in AD patients [16]. In contrast, another study [81] showed that during the early stages of AD, in *APOE4* carriers, the increases in the CSF inflammatory markers were related to preserved cognitive performance, suggesting that the synthesis of Aβ pathology-induced cytokines might be related to early-stage cognitive preservation. These inconsistent findings suggest that the relationship between neuroinflammation and neurocognitive functioning may not be linear in nature. Neuroinflammation is a dynamic process that manifests various effects as the disease progresses [82,83]. At the asymptomatic stage, the microglia could initially help clean the Aβ deposition once they sense it and play their protective roles. However, changes induced by other factors, including genetic susceptibility and especially chronic inflammation, might also interact with the intrinsic immune responses in the brain. With disease progression, the imbalance between Aβ production and clearance will ultimately trigger the exaggerated and uncontrollable microglial activation even without specifically targeted pathogen proteins. Such long-lasting neuroinflammation is deleterious and will finally reduce the synaptic plasticity and be lethal to neurons [19,83,84].

Neuroinflammation is involved in myelin impairment [85]. Essentially, aged animals show weakened remyelination capacity because of the loss of environmental homeostasis and reduced differentiation of oligodendrocyte precursor cells into remyelinating oligodendrocytes [86,87]. More seriously, evidence has shown that microglia and astrocytes might influence the remyelination and demyelination processes directly or indirectly by interacting with oligodendrocytes [88]. Microglia could prune myelin sheaths and modify myelin development [89]. The depletion of microglia results in excessive and ectopic myelin executed by oligodendrocytes [90]. Interestingly, by testing immunoreactivity to microglia and astrocytes in white matter, researchers have found that glial fibrillary acidic protein rather than ionized calcium-binding adaptor molecule 1 is negatively correlated with myelin, indicating that astrocytosis instead of microglia in white matter is associated with loss of myelin, even during normal aging [91]. Therefore, during the aging process, microglia [92] and microglial priming (e.g., increased sensitivity to inflammatory responses [83]) will increase as well as the epigenetic modifications that lead to sustained inflammation in astrocytes [93]. These morphological and functional changes in glial cells increase neurons’ susceptibility to demyelination, which explains age-related neurocognitive decline. This speculation is substantiated by previous research that showed age-related slowing in cognitive processing was significantly associated with demyelination and myelin integrity in healthy older adults [94,95]. Moreover, abnormal white matter microstructure, which has an implied association with neuroinflammation or myelin-related pathologies, contributes to memory impairment [96]. Amnestic MCI participants show alterations of white matter microstructure in the fornix, uncinate fasciculus, and parahippocampal cingulum [97], which have been widely recognized as related to the false recollection of memory [98] and associative and episodic memory functions, respectively [99,100]. In sum, these regional cellular changes might explain the links between chronic neuroinflammation and behavioral abnormalities.

Inflammation could affect the BBB’s properties during aging, leading to neurocognitive changes. The BBB is a specialized structure comprising astrocytes, pericytes, and endothelial cells [101]. It gate-keeps and controls the ionic homeostasis, nutrient, and water balance against pathogens and toxins [102]. However, the BBB changes during aging, especially when challenged by inflammation. In normal aging, with endothelium degeneration, a decreased number of pericytes, and attenuated expression of tight-junction proteins as well as transporter dysfunctions, disruptive (histological level) and nondisruptive (molecular level) changes will occur in the BBB [103]. The BBB’s breakdown leads to leukocyte infiltration and cytokine invasion, which aggravates CNS inflammatory responses [104]. Furthermore, several cytokines (IFN-γ [105], IL-1β, and TNF-α [106]) have been reported to attack the BBB or induce cerebral endothelial activation, causing BBB disruptions. Voirin et al. [107] studied an in vitro BBB model. They observed that inflammatory stress changed the BBB’s properties such that its permeability increased, leading to dysfunction of the tight junctions and ATP-binding cassette transporters. Propson et al. [108] observed that the endothelial C3a receptor mediates vascular inflammation and increases BBB permeability during aging, implicating complementary systems’ critical role in regulating the innate immunity in the brain. Overall, such age-related BBB openings might further induce astrocytic transforming growth factor β, signaling hyperactivation in mice [109], which leads to hippocampal hyperexcitability and aberrant electrocorticographic activity. These abnormal neural signals could be excitotoxic and boost the release of toxic proteins [110], being regarded as early biomarkers of MCI and related to α-synuclein-mediated neurodegeneration [111]. Additionally, human studies have shown that individuals with early cognitive dysfunction presented BBB breakdown in the hippocampus independent from pathological protein aggregations [112]. Finally, researchers have discussed the specific roles of glia (i.e., astrocytes [113], microglia [89], and oligodendrocytes [114]) in inflammation in other comprehensive reviews.

## 5. Organelle Dysfunction and Abnormal Lipid Metabolism

Recently, some studies have discovered organelle dysfunction and abnormal lipid metabolism in the exacerbation of inflammation and neurodegenerative processes. As one of the versatile intracellular organelles, peroxisomes play a role in ROS metabolism by producing and removing H_2_O_2_ as well as modulating lipid biosynthesis and metabolism [115]. Peroxisomes are also involved in regulating inflammation directly and indirectly. During a microbial infection, peroxisomes play essential roles in facilitating cytoskeleton rearrangement and phagocytosis in macrophages of *Drosophila* and mice. Moreover, they regulate H_2_O_2_ and nitric oxide turnover and activate p38-MAPK signaling to initiate innate immune responses in response to infection challenges [116]. Peroxisomes also regulate the immune processes by controlling lipid metabolisms and thereby affect the functions of membrane-localized receptors and proinflammatory signaling [115].

Peroxisomal function becomes frail with aging [117]. Accumulating evidence suggests that peroxisome dysfunctions lead to reduced antioxidants and increased risks of protein, DNA, and lipid oxidation; influence mitochondrial integrity; and cause mitochondrial fragmentation [118], which contributes to cell senescence and perpetuates age-related chronic inflammation, and finally result in the pathogenesis and progression of neurodegenerative changes [119]. Compared to the mouse model lacking multifunctional protein-2 (MFP2) in neurons, astrocytes, and oligodendrocytes but not microglia, in mouse models with deleted MFP2, an essential enzyme in peroxisomal β-oxidation and maintaining lipid homeostasis, earlier and stronger microglial activation, severer neuronal dysfunction, reduced grip strength, fewer exploration behaviors, and reduced life span occur [120].

Continuing the previous discussion on myelinations, it is worth noting that lipids and membrane lipids make up 50–60% of the solid material in the brain [121]. Overall, due to the age-related decline in peroxisomal functions, alterations in redox balance, accumulation of very-long-chain fatty acids, increased cholesterol, and decreased plasminogen and docosahexaenoic acid will occur, which induce the accumulation of toxic proteins such as Aβ, tau, and a-synuclein. This series of changes is detected in AD and Parkinsonism [119]. Moreover, cholesterol is involved in the generation of Aβ. One study showed that apolipoprotein E specifically uses astrocyte-derived cholesterol to transport neuronal APP into and out of lipid clusters, one containing β and γ secretase and one promoting Aβ peptides. Cutting down the cholesterol levels in cultured neurons causes APP to exit lipid clusters and transform into neuroprotective soluble APP-α [122].

Redox imbalance is related to lipid and cholesterol oxidation and oxysterol formations in AD. AD patients experience increased lipid peroxidation products (i.e., malondialdehyde) and reduced antioxidant enzymes (i.e., glutathione peroxidase) in red blood cells and plasma. Furthermore, MMSE scores are reversely correlated with malondialdehyde and with conjugated dienes (an index representing the oxidation of polyunsaturated fatty acids) [123]. Several inflammatory molecules and oxysterol levels have shown corresponding changes along with the various stages of AD. At the later stages of AD, 24-hydroxycholesterol (24-OH) levels sharply decrease, 27-hydroxycholesterol levels double, and levels of other oxysterols increase (i.e., 25-OH, 5β,6β-epoxycholesterol, 4α-OH, and 4β-OH), implicating the roles of cholesterol oxidation and oxidative stress in AD [124]. Interestingly, other studies have shown that one of the oxysterols, 25-hydroxycholesterol, exerts an anti-inflammatory effect by inhibiting IFN-γ receptor trafficking to lipid rafts by disrupting the formation of rafts in microglia [125], suggesting the multifaceted effect of oxysterols.

## 6. Glymphatic System

The glymphatic system is a CSF transport system located in the perivascular space in the brain that is formed by the capillary and vascular end-feet of astrocytes (Figure 2) [126,127]. Produced by the choroid plexus, CSF moves along the artery within the perivascular spaces [128]. One of the glymphatic system’s most important responsibilities is waste clearance [129]. With the facilitation of aquaporin-4 (AQP4) water channels located near the end-feet of astrocytes, CSF will then enter the brain parenchyma and mix with interstitial fluid (ISF) [130]. Thus, interstitial proteins, as well as ISF, will be propelled to leave the brain via perivenous space in this current [131]. This process helps clean brain metabolites, including Aβ in AD [132]. Another role of the glymphatic system is to distribute nutrients and deliver therapeutic agents [126,133]. During inflammation, CSF flow can be impeded, affecting the clearance of protein because of the excessive immune cells in the perivascular spaces [134]. Meanwhile, peripheral cytokine may enter the brain parenchyma by bypassing the arachnoid mater and via the glymphatic system [135]. During aging, although the expression of AQP4 increases, a loss of AQP4 polarization occurs, meaning AQP4 will change location from the perivascular side to the whole membrane. One postmortem study [136] showed that in those who are older than 85 but remain cognitively intact, perivascular AQP4 localization in the frontal cortex is preserved whereas the loss of perivascular AQP4 localization is associated with an increased Aβ burden and increased Braak stage when controlling for age. The study indirectly emphasized the role of the glymphatic clearances with AQP4 as the structural basis in neurodegeneration. Therefore, older people with chronic inflammation are exposed to double jeopardy predisposing and/or precipitating neurocognitive decline.

## 7. Potential Interventions

### 7.1. Pharmacological Approaches

Pharmacological approaches to controlling chronic inflammation may be beneficial to neurocognitive functioning. Nonsteroidal anti-inflammatory drugs (NSAIDs), a group of drugs that inhibit the cyclooxygenase (COX) and then restrict the biosynthesis of proinflammatory prostaglandins, may help reduce ROS, inhibit the NF-kB pathway, and activate the peroxisome proliferator-activated receptor γ that regulates anti-inflammatory responses [137,138]. Several specific mechanisms, such as preventing mitochondrial Ca^2+^ overload and Aβ1-42-induced apoptosis, might be the key factors by which NSAIDs exert their effect on the CNS [139]. Some early studies have shown that long-term use of ibuprofen was associated with a delay in cognitive decline [140]. Several comprehensive reviews and meta-analyses on randomized controlled trials evaluating the beneficial effects of anti-inflammatory drugs on cognition show inconsistent findings (e.g., [20,141,142,143,144,145]). One Cochrane review did not produce evidence supporting the use of low-dose aspirin or other NSAIDs of any class (celecoxib, rofecoxib, or naproxen) for the prevention of dementia, but gastrointestinal adverse events and potential harm were detected [146]. Additionally, no proof supports the efficacy of aspirin, steroids, or NSAIDs (traditional NSAIDs and COX-2 inhibitors) [147]. This result indicates that further experiments are required to confirm the effects of anti-inflammatory drugs and to evaluate their various effects on people in different stages of the aging process.

### 7.2. Lifestyle Management

Nutritional intervention aims to reduce the overall level of inflammation. For example, research has shown that omega-3 long-chain polyunsaturated fatty acids and eicosapentaenoic and docosahexaenoic acids, in combination with vitamins (B complex and D3), phytochemicals (e.g., flavonoids such as resveratrol and polyphenols such as curcumin), alkaloids (e.g., caffeine), probiotics, and short-chain fatty acids (SCFAs; e.g., butyrate) could effectively control inflammation states or directly reduce IL-6 and CRP levels [141,148]. Moreover, some widely recognized beneficial foods, such as blueberries, might also improve cognition by reducing body inflammation [149]. However, simple carbohydrates, saturated and trans-fatty acids, and processed foods could be proinflammatory and should be consumed with caution [148]. Nutritional interventions’ beneficial effects on neurocognitive functions could occur via communication along the brain-gut axis. Gastrointestinal microbiota affect the brain through the synthesis and secretion of metabolites (e.g., bile acids, choline, and SCFAs) and the production of neurotransmitters or precursors (e.g., serotonin and tryptophan), which enter the bloodstream through the microvilli [150,151]. The dysbiosis of gut microbiota is prominent in aging [152]. Dietary and nutritional patterns not only affect the gut microbiota diversity [153] but also alter the patterns of metabolites and regulate epithelial cells’ immune responses [154] as well as the circulating levels of pro- and/or anti-inflammatory cytokines [155]. One study identified the MCI-specific mycobiome signatures [156], suggesting the possibility of modulating food intake habits and microbiota patterns to suppress chronic inflammation and thus improve neurocognitive functions.

Quality sleep can have a significant influence on neurocognitive functioning [157,158,159]. However, age-related sleep disturbance is common [160,161]. Besides the adverse psychological effects lack of sleep causes, its negative effect on glymphatic functioning is also noteworthy [162,163]. Poor sleep affects glymphatic functioning because increased glymphatic activity has been detected during sleep, when the central levels of norepinephrine are relatively low [164]. Fluid exchanges mainly begin and proceed during nonrapid eye movement sleep (NREM) sleep. Therefore, fragmented sleep patterns, characterized by the deficiency of stage 3 NREM sleep; the persistent disturbance of stage 1 and 2 NREM sleep; and shortened total sleep time accompanying aging all contribute to disrupted glymphatic activity [165,166]. Therefore, interventions promoting sleep quality will likely benefit the functioning of the glymphatic system, which in turn helps regulate inflammation.

Physical exercise shows neuroprotective effects by controlling neuroinflammatory status. Specifically, exercise might not only help improve BBB permeability [167] but also increase the glymphatic flow in mice [168]. Moreover, regular exercise could promote cognitive health by altering the levels of cytokines in the peripheral circulation and key brain regions. For instance, among rats, daily runners showed lower levels of hippocampal IL-1β and circulating monocyte chemoattractant protein-1 but higher hippocampal IL-18 concentration [169]. Exercise significantly improves the spatial ability in aging rats via the neuroimmune pathways of increased local IL-18 concentrations in the hippocampus and in neurogenesis [169]. Another study showed that in streptozotocin-induced AD model mice, treadmill exercise helps suppress the production of proinflammatory cytokines in microglia and reduce oxidative stress, inhibits hippocampal neuronal degeneration, and alleviates cognitive deficits [170]. Research on humans supports the interaction effects of exercise and neurotrophic factors on cognitive functioning [171]. In a 12-week randomized active-controlled trial of the therapeutic effect of *wu xing ping heng gong* (qigong) on aging-sensitive neurocognitive abilities, there was significant improvement in neurocognitive abilities, increased hippocampal volume, and reduced peripheral IL-6 levels. Moreover, following qigong training, a greater reduction in peripheral IL-6 levels was associated with a greater increase in processing speed and a more significant training-induced effect of hippocampal volume on improvement in sustained attention [42].

Other potential interventions involved targeting a specific clearance of senescent cells peripherally and in the CNS [168,172]. This approach aims to minimize the harm caused by the uncontrollable activation of microglia and senescence of immune cells. However, it is worth carefully considering which cells we should target and the effects such targeting will have. For example, depleting senescent microglia [173,174] helps alleviate inflammation, but it is detrimental to cognition with the elimination of the senescent neuroblasts by natural killer cells during normal brain aging [175,176]. Other reviews proposing theoretical frameworks and potential therapeutic interventions based on inflammation are listed here, including vascular inflammation [177], immune cell migration [178], and inflammasomes [179].

## 8. Discussion

We have reviewed aging’s effects on the immune system and neurocognition. Furthermore, we have explored the potential mechanisms of how and why such immune-neural communications could occur. We have also discussed several physiological changes in vascular and blood barriers in the brain, activations of resident brain cells, and the special role of glymphatic systems in maintaining a healthy brain environment. By synthesizing the current evidence of the roles of some key cytokines, we provide an overview of this topic. Nevertheless, contradictions, limitations, and ambiguities also exist among the evidence that are worthy of attention. Here, we present and discuss future directions.

First, the operational definitions of chronic inflammation and its measurements in animals and people as well as in research and clinical contexts require further clarification. The Centers for Disease Control and Prevention, in collaboration with the American Heart Association, has recommended the following criteria for identifying inflammation risk based on the average level of CRP measured in fasting and not fasting: low risk: <1.0 mg/L, average risk: 1.0 to 3.0 mg/L, and high risk: >3.0 mg/L [180]. CRP is also widely used in the context of aging research [181]. However, due to cytokines’ and chemokines’ properties and the corresponding requirements for assay techniques and analyte stability, the clinical applications of cytokines are constrained. Moreover, characterizing the cytokines and other related inflammatory markers in the context of chronic inflammation is still necessary to gain an in-depth and specific understanding of the immune mechanisms. Currently, it is hard to determine the “best” inflammatory markers to characterize normal cognitive aging in healthy older adults and to identify or differentiate between populations with healthy aging, amnestic and nonamnestic MCI, and various types of dementia, mainly because of cytokine markers’ unsatisfactory stability, sensitivity, and specificity. For example, higher levels of IL-1β and IL-12 are associated with nonamnestic multiple-domain MCI [182]. However, another meta-analysis showed that only IL-1β, rather than IL-6, TNF-α, and CRP, was significantly elevated in AD [183]. The CRP and IL-6 were associated with all-cause dementia rather than AD [15]. Therefore, to resolve the heterogeneity of research methodologies and facilitate the accumulation of solid and generalizable evidence, future researchers should better consider the above-mentioned age-related changes in inflammatory markers, reevaluate and select the most sensitive and reliable markers, and utilize average values of data from multiple time points to optimize the characterization of chronic inflammation.

Second, the biological and psychological roles of inflammatory mediators (i.e., cytokines and chemokines) need more elucidation. It is obvious that in animal studies, the effects of cytokines range from immune aspects to basic neural-activity aspects and generally show beneficial and detrimental effects. Although we should consider the variability of research methodology, it also suggests that chronic inflammation and immune dysfunction do not influence the brain and cognitive functions in a monotonic manner. Therefore, at the molecular and behavioral levels, we need further evidence that clarifies this multifaceted relationship: What factors determine whether inflammation is benign or malignant for cognitive functions? Does inflammation play multiple roles (helpful or harmful) in normal cognitive aging and pathological states?

Third, the bidirectional relationship between chronic inflammation and brain degeneration can be a problematic issue in interventions. The disruption of the BBB, neuroinflammation, glymphatic system dysfunctions, and pathological protein aggregation form a vicious circle. During aging, these processes interact with and exacerbate each other. More specifically, besides the effect of a disruption of the BBB on the microglia and astrocytes [104], pericytes, a basic component of cerebral capillaries, also constantly regulate neuroinflammation [184]. However, microglia can maintain BBB integrity by expressing claudin-5 and approaching the endothelial cells at an early stage [185] or by down-regulating the levels of protein phosphorylation and phagocytic vesicles as well as the role of astrocytes in repairing the BBB [186]. However, during prolonged inflammation states, microglia become aggressive, phagocytosing astrocytic end-feet [185] and producing ROS through NADPH oxidase [187], which could be detrimental to the structural stability of the BBB and glymphatic systems. Therefore, could preventing the deterioration of one session break down the whole cycle? Which part would be easier to regulate or implement for the intervention? These questions await answers.

Fourth, translational and human research explaining molecular-neural-behavioral links is still lacking. Human and animal studies have shown that chronic inflammation affects global cognition and specific cognitive domains [13,188,189,190]. Although we have collected information on inflammation’s mediating role in explaining the relationship between chronological age and cognitive dysfunctions [14], the intermediate linkers are missing. More specifically, one of the current limitations is that most of the experiments testing causality have been conducted in animal models, which significantly limits the study of prefrontal-lobe-related changes to a higher level of function. However, longitudinal data delineating causality and temporal changes in cognitive functions are currently rare. One longitudinal study demonstrated that elevated inflammation before or during middle adulthood predicts greater white matter hyperintensity volume and reduced white matter microstructural integrity in individuals when they are older [191], bringing together the pieces of the puzzle. However, many issues remain unresolved: Which cognitive domain is the most vulnerable to inflammation? How does it relate to the cognitive-aging trajectory? Do changes in brain structure [192] and function [193] mediate the relationship between chronic inflammation and cognitive changes? To what extent or at which stage does inflammation drive the shift from normal cognitive aging to pathological changes? Future researchers could invest more time and energy in conducting longitudinal studies on human participants with neuroimaging tools to validate these relationships and provide clinical guidance.

In terms of interventional trials in older people, individual lifestyle characteristics and medical histories should be considered in the study design and measurement.

## 9. Conclusions

Age-related neurocognitive declines leading to the onset of MCI or AD severely affect people’s quality of life and independence. We reviewed the relationships between chronic inflammation and cognitive aging and the potential biological mechanisms underpinning these relationships. Knowledge of the relationships between chronic inflammation and neurocognitive functioning as we age offers significant insight into the pathophysiological mechanisms underpinning normal and pathological aging processes. Furthermore, pharmacotherapy and/or behavioral interventions to promote healthy immune responses against inflammation may buffer age-related neurocognitive decline and neurodegeneration.

## Figures and Tables

**Figure 1 ijms-23-12573-f001:**
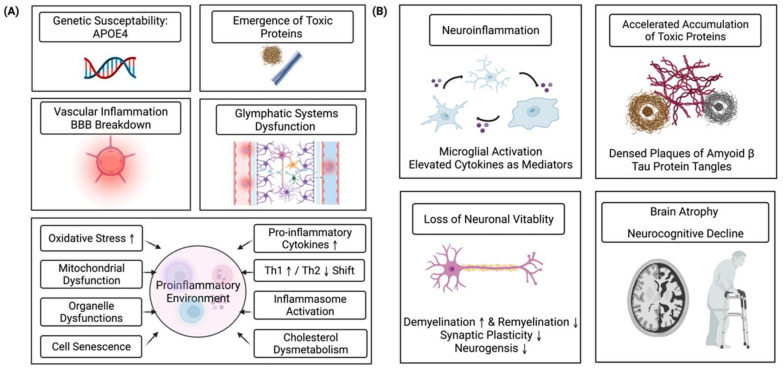
The proposed mechanisms underpinning the relationship between chronic inflammation and neurocognitive decline. Panel (**A**) shows the factors initiating or precipitating neuroinflammation and cognitive decline, including genetic susceptibility, vascular inflammation and blood-brain barrier (BBB) breakdown, the emergence of toxic proteins, glymphatic dysfunctions, and chronic inflammation in the aging process. Panel (**B**) presents degenerative changes in the brain due to inflammation and elevated cytokines. Microglial activation with morphological and functional changes is a key event in neuroinflammation. Elevated cytokines affect neural activities and mediate environmental homeostasis. With the accelerated production and deposition of neuropathological proteins, neurons gradually lose their vitality due to the loss of synaptic plasticity, reduced neurogenesis, and demyelination. Finally, researchers have detected microstructural changes in white matter and brain atrophy. Older people generally show behavioral changes, including slower processing speed, memory loss, and declines in executive functions. We created this figure using BioRender.com, accessed on 19 July 2022.

**Figure 2 ijms-23-12573-f002:**
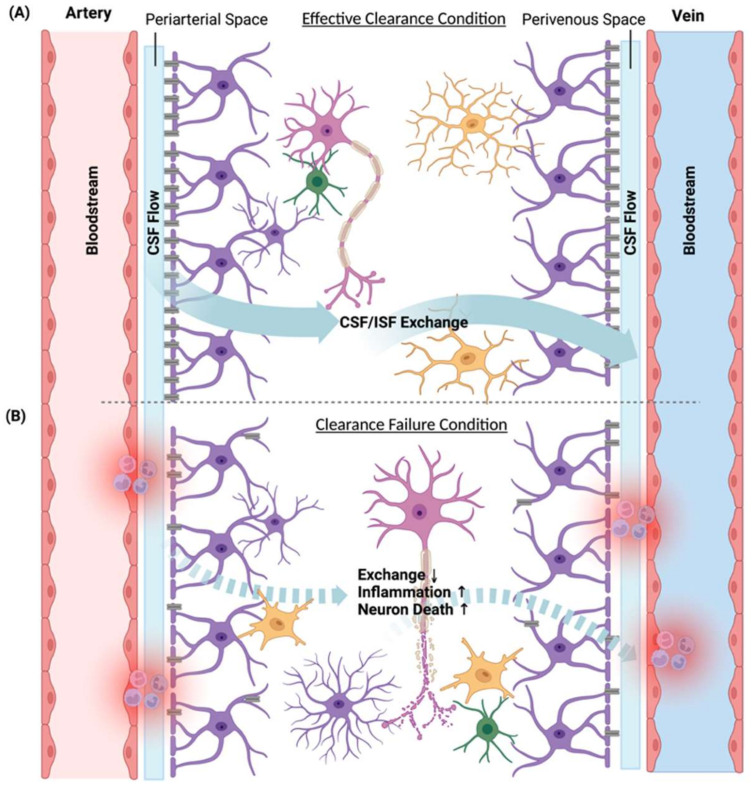
The glymphatic system. Structurally, the glymphatic space is a composite of the periarterial and perivenous space, surrounded by the blood vessel walls and vascular end-feet of astrocytes (in purple). Within the glymphatic spaces, arterial pulsations drive the flow of the cerebrospinal fluid (CSF), and the fluid enters the brain parenchyma via aquaporin-4 (AQP4) water channels. Lastly, this current will converge into the perivenous space. The exchanges between CSF and interstitial fluid (ISF) are particularly important for material exchanges and waste cleaning in the brain. Panel (**A**) shows effective cleaning of the metabolic waste in the glymphatic system. Neurons (in pink), microglia (in yellow), and astrocytes and oligodendrocytes (in green) are in normal conditions. In panel (**B**), during aging and chronic inflammation, a significant reduction occurs in the CSF/ISF flow exchanges due to age-related AQP4 depolarization. Furthermore, excessive immune cells in the perivascular spaces as well as sleep disturbances adversely affect the functioning of the glymphatic system, resulting in neuronal death and altered gene expression profiles in glial cells. We created this figure using BioRender.com, accessed on 19 July 2022.

## Data Availability

Not applicable.

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
