# Peer review of "Relationships between Inflammation and Age-Related Neurocognitive Changes"

_ijms, 2022, doi:10.3390/ijms232012573_

Round 1
Reviewer 1 Report
This is an interesting and somewhat well-written review on the relationship between inflammation and neurocognitive changes in the elderly. The authors have succinctly outlined some of the important mediators of inflammation and have attempted to associate it with cognitive changes in humans and humanized models of neurodegenerative diseases. This is a very large field of study and the authors should acknowledge that the scope of their review has been limited to a very select group of publications. They should refer the reader to more comprehensive reviews that are available for each of the inflammatory markers, glial and immune cells and potential therapeutic interventions.
Comments
- Ln 31 - Oxidative stress and mitochondrial dysfunction should be briefly introduced
- Ln 37-38 – in what was increased IL-6 measured? Blood, CSF?
- Ln 45 – what is CRP? An explanation of its function should be provided in order for the reader to appreciate why and increase is detrimental.
- Ln 47-48 – incomplete sentence – “…..could be triggered by and in …..” Could be triggered by what?
- A brief introduction to the interleukins should be provided at the start of Section 3.
- Section 3 must be divided into sub-sections with headers to guide the reader
- Ln 64 – define CNS
- Ln 100 – APP is not deposited in AD, it is Ab that is deposited as plaques.
- Ln 115 – this paragraph requires an introduction to IL-12 and its subunits prior to the information on the role(s) of IL-12.
- Ln 175 – an introduction to interferons is also warranted.
- There is no in-text reference to either Figure 1 or Figure 2.
- The manuscript must a be reviewed by a native speaker as some of the grammar requires correction.
Reviewer 2 Report
The review is well written but only present conventionnal aspects.
Two additionnal aspects could be added in Figure 1: organelle dysfunctions and perurbation of lipid metabolism (fatty acids and cholesterol)
In ageing, there are also organelle dysfunctions in the brain and this can have potential impacts on brain inflammation. This is mainly the case of peroxisomal dyfunctions which are involved in neurodegeneration. You can cite the following papers (Di Cara F, Andreoletti P, Trompier D, Vejux A, Bülow MH, Sellin J, Lizard G, Cherkaoui-Malki M, Savary S.
Peroxisomes in Immune Response and Inflammation. Int J Mol Sci. 2019 Aug 8;20(16):3877. doi: 10.3390/ijms20163877. ) and also add paper from the teams of Marc Fransen Meryam Baes. Don't forget that peroxisome and mitochondria are functionnaly tightly connected. In addition, peroxysome is invoved in the degradation of eicosanoids and this non cytokinic nflammation can also favor brain inflammation. Papers on mitochondrial dysfunction and inflammation must also be added. This aspect on organelle dysfunction in the brain, its impact on inflammation and neurodegeneration must be developped. This will greatly improve the review which will be more attractive.
In addition, the brain is very rich in lipids especially in cholesterol. Most of the brain cholesterol is located in the brain. Under the action of oxidative stress, cholesterol in the brain is transformed in oxysterols which are strong pro-inflammatory molecules. Some papers must be added such as (Zarrouk A, Hammouda S, Ghzaiel I, Hammami S, Khamlaoui W, Ahmed SH, Lizard G, Hammami M. Association Between Oxidative Stress and Altered Cholesterol Metabolism in Alzheimer's Disease Patients. Curr Alzheimer Res. 2020;17(9):823-834. doi: 10.2174/1567205017666201203123046;
Testa G, Staurenghi E, Zerbinati C, Gargiulo S, Iuliano L, Giaccone G, Fantò F, Poli G, Leonarduzzi G, Gamba P. Changes in brain oxysterols at different stages of Alzheimer's disease: Their involvement in neuroinflammation. Redox Biol. 2016 Dec;10:24-33. doi: 10.1016/j.redox.2016.09.001.) and this point must also be developped. Oxysterols present at increased levels in the brain of Alzheimer patients are known as strong inducers of inflammation. The relationship between cholesterol, cholesterol metabolism, oxidative stress and inflammation in neurodegeneration is very important. The authors must also develop this point.
The two remarks required important modifications. The new paragraphs will improve the originality of the review and its attractivity.
Round 2
Reviewer 1 Report
We thank the authors for considering the comments made by the reviewers and incorporating the indicated changes.
The language has been improved but still requires work by a Language Editing service or the Journals' editorial team.
Author Response
Thank you for Reviewer 1's comment. Please be informed that we have solicited a professional language editing service to help improve the language of our manuscript.
Reviewer 2 Report
This is an objective review on neuroinflammation in neurocognitive changes.
Different aspects have been treated including how to evaluate neuroinflammation in patients.
The paper is clear, well written and illustrated.
Author Response
Thank you Reviewer 2 for the time and effort spent on our paper.